# Dendritic Cell-Triggered Immune Activation Goes along with Provision of (Leukemia-Specific) Integrin Beta 7-Expressing Immune Cells and Improved Antileukemic Processes

**DOI:** 10.3390/ijms24010463

**Published:** 2022-12-27

**Authors:** Elias Rackl, Lin Li, Lara Kristina Klauer, Selda Ugur, Elena Pepeldjiyska, Corinna L. Seidel, Carina Gunsilius, Melanie Weinmann, Fatemeh Doraneh-Gard, Nina Reiter, Caroline Plett, Daniel Christoph Amberger, Peter Bojko, Doris Kraemer, Jörg Schmohl, Andreas Rank, Christoph Schmid, Helga Maria Schmetzer

**Affiliations:** 1Department of Medicine III, University Hospital of Munich, 81377 Munich, Germany; 2Department of Orthodontics and Dentofacial Orthopedics, University Hospital, LMU Munich, 80336 Munich, Germany; 3Department of Hematology and Oncology, Rotkreuzklinikum Munich, 80634 Munich, Germany; 4Department of Hematology and Oncology, St.-Josefs-Hospital, 58097 Hagen, Germany; 5Department of Hematology and Oncology, Diakonieklinikum Stuttgart, 70176 Stuttgart, Germany; 6Department of Hematology and Oncology, University Hospital of Augsburg, 86156 Augsburg, Germany

**Keywords:** integrin beta 7, leukemia-derived dendritic cells, immune therapy, AML, MDS

## Abstract

Integrin beta 7 (β7), a subunit of the integrin receptor, is expressed on the surface of immune cells and mediates cell–cell adhesions and interactions, e.g., antitumor or autoimmune reactions. Here, we analyzed, whether the stimulation of immune cells by dendritic cells (of leukemic derivation in AML patients or of monocyte derivation in healthy donors) leads to increased/leukemia-specific β7 expression in immune cells after T-cell-enriched mixed lymphocyte culture—finally leading to improved antileukemic cytotoxicity. Healthy, as well as AML and MDS patients’ whole blood (WB) was treated with Kit-M (granulocyte–macrophage colony-stimulating factor (GM-CSF) + prostaglandin E1 (PGE1)) or Kit-I (GM-CSF + Picibanil) in order to generate DCs (DC_leu_ or monocyte-derived DC), which were then used as stimulator cells in MLC. To quantify antigen/leukemia-specific/antileukemic functionality, a degranulation assay (DEG), an intracellular cytokine assay (INTCYT) and a cytotoxicity fluorolysis assay (CTX) were used. (Leukemia-specific) cell subtypes were quantified via flow cytometry. The Kit treatment of WB (compared to the control) resulted in the generation of DC/DC_leu_, which induced increased activation of innate and adaptive cells after MLC. Kit-pretreated WB (vs. the control) led to significantly increased frequencies of β7-expressing T-cells, degranulating and intracellular cytokine-producing β7-expressing immune cells and, in patients’ samples, increased blast lysis. Positive correlations were found between the Kit-M-mediated improvement of blast lysis (vs. the control) and frequencies of β7-expressing T-cells. Our findings indicate that DC-based immune therapies might be able to specifically activate the immune system against blasts going along with increased frequencies of (leukemia-specific) β7-expressing immune cells. Furthermore, β7 might qualify as a predictor for the efficiency and the success of AML and/or MDS therapies.

## 1. Introduction

### 1.1. Acute Myeloid Leukemia (AML) and Myelodysplastic Syndrome (MDS)

AML and MDS are clonal stem cell disorders of hematopoiesis that lead to the uncontrolled proliferation of progenitor cells and the suppression of healthy, functional cells [1,2], going along with infections, bleeding, thrombosis and anemia [3]. Their prognosis depends on chromosomal aberrations, blast counts and patients’ ages. Standard treatment for AML/high-grade MDS is based on high-dose chemotherapy, hypomethylating agents and stem cell transplantation (SCT), leading to high rates of remission (66–90% of AML cases after 1–2 therapy cycles), however high rates of relapse have also been noted in up to 80% of cases in the following two years [4,5,6].

### 1.2. DC-Based Immunotherapeutic Approaches

DCs mediate immune responses via major-histocompatibility-complex (MHC)-associated antigen-presentation [7]. In vivo DCs derive from hematopoietic stem cells and monocytes and undergo an activation and maturation process in conjunction with the upregulation of cell adhesion receptors, chemokine receptors (e.g., CCR7), MHC antigens and other co-stimulatory factors [8,9,10].

DCs can be generated ex vivo from healthy monocytes or from myeloid blasts in patients with myeloid leukemia (DC_leu_, leukemia-derived DC-expressing individual patients’ leukemic antigens). DC/DC_leu_ and their subtypes, e.g., DC_leu_, mature DC (DC_mat_) and mature leukemia-derived DC (DC_leu-mat_) (Table 1), can be generated from whole blood (WB) using selected combinations of response modifiers (e.g., GM-CSF (granulocyte macrophage colony-stimulating factor) + PGE1 (prostaglandin E1) (Kit-M) or GM-CSF+Picibanil (Kit-I)) without the induction of blast proliferation (as detected by blasts’ co-expression of transferrin receptor CD71 (Bla_prol-CD71_) or intracellular proliferation marker IPO38 (Bla_prol-IPO38_); see Table 1) [10,11,12,13].

### 1.3. Key Players of Immune Defense and Their Detection

Immune defense is mediated by humoral and cellular mechanisms of the innate and adaptive immune system [22,23]. The key players of the (fast-acting) innate immune system are monocytes (CD14^+^), macrophages (CD15^+^), DCs (e.g., CD206^+^, CD80^+^, etc.) as well as NK-cells (CD56^+^CD3^−^) and CIK-cells (CD56^+^CD3^+^) [13,24,25,26]. The key players of the (antigen-specific/long-lasting) adaptive immune system are B-cells (CD19^+^), T-cells (CD3^+^) and their subsets: naive or non-naive T-cells (T_naive_, CD3^+^CD45RO^−^; T_non-naive_, CD3^+^CD45RO^+^; T_non-naiveCD4+_, CD4^+^CD3^+^CD45RO^+^; and T_non-naiveCD4−_, CD4^−^CD3^+^CD45RO^+^), which give rise to central memory cells (T_cm_, CD3^+^CD45RO^+^CCR7^+^), enabling faster reactivation of the immune system against reoccurring targets (see Table 1) [10,27,28]. These cells can be detected via flow cytometry.

Leukemia-specific/antileukemic cells can be quantified using functional analyses. The degranulation assay (DEG) detects the CD107a molecule left on the cell surface after the release of granzymes and perforins [29,30]. The intracellular cytokine assay (INTCYT) allows the intracellular quantification of cytokines (interferon gamma (IFNγ) and tumor necrosis factor alpha (TNFα)), which are considered specific triggers of the immune responses and mediators of cell apoptosis [21,30,31]. Antileukemic blast lytic effects can be detected, e.g., using a non-radioactive fluorolysis assay [13,32,33].

### 1.4. Integrin Beta 7 (β7)

The β7 subunit, expressed on adaptive and innate immune cells, plays an important role in cell–cell-adhesion [15,34,35] (e.g., β7-expressing T-cells (T_β7+_), T_non-naive_ (T_non-naiveβ7+_), T_naive_ (T_naiveβ7+_), T_cm_ (T_cmβ7+_), NK-cells (NK_β7+_)- and CIK-cells (CIK_β7+_); see Table 1). β7 expressed in hematopoietic stem cells (HSC) plays a role in HSC homing [36]. β7 is primarily known for its function in T-cell-trafficking to the gut via interaction with mucosal-addressin-cell-adhesion-molecule-1 (MAdCAM-1) [35,37]. Higher β7 expression on T-cells seems to correlate with cytotoxic effects towards intestinal cells in inflammatory bowel diseases [38] and in NK-cells with higher cytotoxicity in immunodeficiency-virus-infected macaques [39]. β7 has been associated with the intraepithelial differentiation of cytotoxic as well as regulatory T-cells, and therefore, with pro- and anti-inflammatory functions [40]. In the past, higher β7 expression in T-cells could be correlated with higher antileukemic, blastolytic potential in AML samples [15].

### 1.5. Aim of This Study

The aim of this study was to further explore the role of β7 expression in immune cell subpopulations in uncultured peripheral WB or after mixed lymphocyte culture (MLC) of patients’ or healthy donors’ T-cell-enriched cells with Kit-pretreated WB as stimulator cells.

In detail, we explored:DC subpopulations using (Kit-treated vs. untreated) WB from leukemia patients (or healthy individuals);(β7-expressing) innate and adaptive immune cells before and after MLC with Kit-pretreated vs. untreated WB from leukemia patients (or healthy individuals);The antileukemic activity of cells using the cytotoxicity fluorolysis assay (CTX) after MLC of patients’ T-cells with Kit-pretreated vs. untreated WB as stimulator cells;Leukemia-specific cells using DEG and INCYT assays in uncultivated WB from leukemia patients (vs. comparable cells for healthy individuals) and after MLC with Kit-pretreated vs. untreated WB;The correlations between antileukemic functionality, leukemia-specific activity and (β7-expressing) immune cells;The correlations between patients’ clinical outcomes/prognostic risk assessment and (β7-expressing) immune cells.

## 2. Results

We further explored the role of β7-expressing (leukemia-specific) immune-reactive cell populations as prognostic markers to predict the clinical outcome, as well as to mediate antileukemic functionality after stimulation. Therefore, we studied the expression of β7 on uncultured immune-reactive cells from AML patients and healthy blood donors (as a control), generated DC/DC_leu_ from healthy and leukemic samples and studied their potential to activate (β7-expressing) immune-reactive cells after T-cell-enriched MLC.

### 2.1. Generation of DC (Subtypes) from WB

#### 2.1.1. Significantly Higher Frequencies of DCs and Their Subtypes in Patients’ and Healthy WB Samples after Kit Treatment Compared to Control (without Added Kits)

We found significantly higher frequencies of DC/DC_leu_ subtypes under the influence of Kit-M (DC(M)) or Kit-I (DC(I) compared to the control (DC(C) in healthy or AML patients’ samples (e.g., %DC/cells, Figure 1A1,B1). Moreover, in AML samples, frequencies of (mature) DC_leu_ were significantly increased compared to the control (Figure 1A1,A2).

#### 2.1.2. No Influence of Kit Treatment on Proliferation of Blasts or Monocytes

The frequencies of proliferating blasts (Bla_prol-CD71_, Bla_prol-IPO38_; Figure 1A1) were comparable in Kit-treated patients’ WB and that of the control. Moreover, the frequencies of proliferating monocytes (Mon_prol-CD71_; Figure 1B2) were comparable in Kit-treated healthy donors’ WB and that of the control.

In summary, we found higher frequencies of DCs and their subtypes (DC_leu_, DC_mat_ and DC_leu-mat_) in Kit-treated WB when compared to the control. The (DC-independent) proliferation of blasts and monocytes was not induced.

#### 2.1.3. Profiles of Immune-Reactive (and Especially β7-Expressing) Cells in Uncultured WB from AML vs. Healthy Blood Donors

Low frequencies of proliferating T-cells (T_prol-early_/CD3^+^, T_prol-late_/CD3^+^), T_cm_/CD3^+^ and innate immune cells were found in uncultured AML as well as in healthy samples. Significantly higher frequencies of NK-cells were found in (uncultured) healthy vs. AML WB samples (Figure 2, MLC(UC)).

Between 18 and 22% of T-cell (subtypes, e.g., T_naive_/T_nonnaive_/T_cm_) and even higher frequencies of innate cells co-expressed β7. The differences in β7 expression were significantly lower in uncultured leukemic vs. healthy T-cells (Figure 2A vs. Figure 2B, MLC(UC)).

Comparing the expression profiles before vs. after T-cell-enriched MLC (using Kit pretreated (or untreated) patients’ or healthy donors’ WB as stimulator cells), we found higher frequencies of activated/proliferating/CD4^−^ T-cells (e.g., T_non-naive_/CD3^+^ (and T_nonnaiveCD4+_/T_CD4+_ and T_non-naiveCD4−_/T_CD4−_) and T_CD4−_/CD3^+^) within patients’ and healthy samples, as well as higher frequencies of CIK/cells and NK/cells (within patients’ samples) after MLC compared to before (Figure 2). Higher frequencies of β7-expressing immune-reactive cell subpopulations (except for T_naiveβ7+_) were found after MLC compared to before (Figure 3).

### 2.2. T-Cell-Enriched Mixed Lymphocyte Culture with Patients’ or Healthy Donors’ Kit-Pretreated (vs. Untreated) WB

#### 2.2.1. Significant Activation and Provision of T-Cells after MLC of Patients’ WB, but Not in Healthy Samples with Kit-Pretreated (vs. Untreated) WB

Comparing the influence of Kit treatment (vs. without) on the composition of immune-reactive cells in patients’ samples, we found significantly higher frequencies of T_non-naive_ and T_cm_ after MLC(M) compared to MLC(CC) (e.g., %T_non-naive_/CD3^+^: MLC(M) vs. MLC(CC), *p* ≤ 0.05; and %T_cm_/CD3^+^: MLC(M) vs. MLC(CC), *p* ≤ 0.05), whereas the frequencies of innate cells were not significantly different (Figure 2A).

No significantly different frequencies of T- or innate cells were found after MLC(M) or MLC(I) vs. the control (MLC(CC) in healthy samples) (Figure 2B).

#### 2.2.2. Significantly Increased Provision of β7-Expressing Immune-Reactive Cells after MLC with Kit-Pretreated Patients’ or Healthy WB Compared to Control (MLC(CC))

After MLC of Kit-treated (vs. untreated) patients’ WB, we found higher frequencies of β7-expressing T-cell subpopulations after MLC(M), e.g., β7-expressing T_cm_ in the T_cm_ fraction (% T_cmβ7+_/T_cm_: MLC(M): 36.58 ± 22.57, *p* ≤ 0.05; MLC(CC): 32.74 ± 19.72)). The frequencies of β7-expressing T-cell subpopulations were comparable after MLC(M) and MLC(I). No differences between MLC(M), MLC(I) and MLC(CC) were found in β7-expressing innate immune-reactive cells (Figure 3A).

After MLC of Kit-treated (vs. untreated) healthy donors’ WB, we found significantly higher frequencies of T_β7+_ and T_cmβ7+_ after MLC(M) and MLC(I) and significantly increased frequencies of T_β7+/_CD3^+^ compared to MLC(CC) (e.g., %T_β7+_/CD3^+^: MLC(M): 33.56 ± 11.62, *p* ≤ 0.05; MLC(CC): 28.54 ± 11.41). The frequencies of β7-expressing cells after MLC(M) and MLC(I) were comparable. No differences could be found in β7-expressing innate immune cells (Figure 3B).

In healthy (compared to patients’) samples, we found significantly higher frequencies of T_β7+_ cells with MLC(UC), MLC(CC) and MLC(M) (*p* ≤ 0.05, Figure 3).

In summary, Kit-treated (vs. untreated) patients’ WB led to higher frequencies of β7-expressing T-cell subtypes, with an induction of non-naive and memory β7-expressing T-cells after MLC. Higher frequencies of β7-expressing T-cell subtypes were found in healthy donors’ (vs. patients’) samples.

#### 2.2.3. Detection of Antigen-Specific (Degranulating or Intracellularly IFNγ-Producing) β7^+^ Immune-Reactive Cells in Uncultured WB from AML and Healthy WB Donors or in Immune-Reactive Cells after T-Cell-Enriched MLC

We found low frequencies of antigen-specific degranulating or intracellularly IFNγ-producing immune-reactive cells in uncultured WB from AML and healthy WB donors. Significantly higher frequencies of T_β7 +IFNγ+_/T_β7+_ were found in uncultured healthy WB samples compared to AML WB samples (Figure 4). (Non-significantly) higher frequencies of these antigen-specific cells were found after stimulation with LAA (in AML samples) and with SEB (in healthy samples). Here, we present data without antigen stimulation.

We found significantly increased frequencies in most of antigen-specific (TNFα- or IFNγ-producing or degranulating) T-cell or innate β7-expressing cell types before vs. after MLC (using Kit pretreated (vs. untreated) patients’ or healthy donors’ WB as stimulator cells (e.g., T_β7 +IFNγ+_/T_β7+_ in WB vs. MLC(CC) or WB vs. MLC(M)). This effect was especially evident within patients’ samples, but was also found in healthy samples.

We found significantly higher frequencies of T_β7+107a+_ and T_β7+IFNγ+_ after MLC(M) compared to MLC(CC) in patients’ samples without LAA stimulation (%T_β7+107a+_/T_β7+_: MLC(M): 60.62 ± 29.62, *p* ≤ 0.05; MLC(CC): 47.15 ± 28.39; T_β7 +IFNγ+_/T_β7+_: MLC(M): 55.24 ± 31.35, *p* ≤ 0.05; MLC(CC): 50.83 ± 33.50). The frequencies of T_β7+TNFα+_ and of NK_β7+107a+_ and CIK_β7+107a+_ were comparable after MLC(M) and MLC(CC) (Figure 4A). In patients’ samples with additional LAA stimulation, the degranulation and intracellular cytokine production after MLC(M) and MLC(CC) were comparable (data not shown).

We were able to find (non-significantly) higher frequencies of T_β7+107a+_ and NK_β7+107a+_ after MLC(M) compared to MLC(CC) in healthy samples without SEB stimulation (Figure 4B). In healthy samples with additional SEB stimulation, the degranulation and intracellular cytokine production of cells after MLC were comparable (data not shown).

We found comparable frequencies of degranulating or cytokine-producing β7^+^ immune cells in healthy compared to patients’ samples (Figure 4A vs. Figure 4B).

In summary, we found an induction of degranulating and intracellular cytokine-producing β7^+^ T-, NK and CIK-cells after MLC using Kit-pretreated WB compared to the control. We also found more degranulation and intracellular cytokine production in healthy donors’ (vs. patients’) samples.

#### 2.2.4. Improved Antileukemic and Blastolytic Functionality of Immune Cells after T-Cell-Enriched MLC with Kit-M-Pretreated (vs. Untreated) WB

We compared the achieved blast lysis (‘lysis’) of MLC(CC), MLC(I) and MLC(M) using the cytotoxicity fluorolysis assay after the co-cultivation of effector (T-cell-enriched MLC stimulated with Kit-pretreated or untreated WB) and target (blast-containing MNC) cells. We quantified the frequencies of viable blasts after 3 and 24 h of incubation time and selected the superior antileukemic effectivity after either 3 or 24 h incubation time as a ‘best of’ value.

After 3 (24) h of incubation of effector with target cells, blast lysis was found in 74.07 (70.37)% of cases after MLC(M) compared to 57.14 (57.14)% after MLC(CC) and 45.45 (54.55)% after MLC(I) (Figure 5A1), with comparable frequencies of blast lysis after MLC(M), MLC(I) and MLC(CC) in cases with achieved lysis (Figure 5A2). After 3 h of incubation of effector with target cells, we found nearly significantly more cases with lysis after MLC(M) compared to MLC(CC) ((*p* = 0.059). Figure 5A1). Improved lysis (compared to MLC(CC)) was found in 62.96 (59.26) % of cases after MLC(M) vs. in 36.36 (45.45) % after MLC(I) (Figure 5B1), going along with comparable frequencies after MLC(M) and MLC(I) (compared to MLC(CC)) in cases with improved lysis (Figure 5B2).

Selecting the best achieved lysis after 3 and 24 h of incubation of effector with target cells (best), we found nearly significantly more cases with lysis after MLC(M) compared to MLC(CC) ((*p* = 0.059). Figure 5A1). Frequencies of lysed blasts after MLC(M), MLC(I) and MLC(CC) in cases with achieved lysis were not significantly different (Figure 5A2). After MLC(M) vs. MLC(I) (compared to MLC(CC)), blast lysis could be improved in 85.19% of cases after MLC(M) and in 72.73% of cases after MLC(I) (Figure 5B1), going along with the comparable frequencies of lysis improvement after MLC(M) and MLC(I) (compared to MLC(CC)) in cases with improved lysis (Figure 5B2).

In summary, Kit-M treatment indicates clearly (although not significantly) improved blast lysis after MLC when compared to the control in our patient cohort.

### 2.3. Correlation of (Antigen-Specific) β7 Expression with Patients’ Allocation to Risk Groups, Response to Induction Chemotherapy and Achieved Antileukemic (Ex Vivo) Functionality

We compared the β7 expression in T-cells and their subtypes in uncultured immune cells in the samples from AML patients at first diagnosis with allocation to the ELN risk groups and response to induction chemotherapy. Patients who achieved (n = 6, vs. those who did not achieve (n = 9)) remission were characterized by clear, although non-significantly higher frequencies of T_β7+_ cells. In patients with favorable (n = 7) vs. adverse ELN risk stratification (n = 3), clear, although non-significantly higher frequencies of T_cmβ7+/_T_cm_ were found (Figure 6A,B).

We correlated the degranulation and intracellular cytokine production (frequencies of (leukemia-specific) β7-expressing immune-reactive cells) with (improved) antileukemic functionality after MLC. We found a clear (although not significant) positive correlation between T_β7+_/CD3^+^ after Kit-M pretreatment and improved blast lysis in MLC(M) (but not in MLC(I)) (compared to MLC(CC)) (r = 0.370; *p* = 0.083) (Figure 7A). We found significant, positive correlations between the frequencies of β7-expressing cells and the frequencies of intracellular cytokine-producing β7-expressing cells (T_β7+IFNγ+_: r = 0.988, *p* < 0.001; T_β7+TNFα+_: r = 0.952, *p* < 0.001) in MLC(M) (Figure 7B). We also found a positive correlation between increased frequencies of T_β7+107a+_ and lysis improvement (r = 0.821; *p* = 0.023) (Figure 7C) with the blastolytic functionality and frequencies of β7-expressing cells.

In summary, Kit treatment of WB led to the increased generation of DC subtypes (DC, DC_leu_, DC_mat_ and DC_leu-mat_), to activated immune cells and to increased frequencies of β7-expressing cells compared to the control. Additionally, Kit treatment led to increased frequencies of degranulating and intracellular cytokine-producing β7-expressing immune cell (subtypes). Finally, (partly significant) we saw positive correlations of β7-expressing immune cell (subtypes) with the provision of leukemia-specific/antileukemic cells after T-cell-enriched MLC, with patients’ responses to chemotherapy or with their allocations to risk types in our small patient cohort.

## 3. Discussion

Expressions of several leukemia-associated antigens (e.g., CD318 and CD11b) have been studied via flow cytometry to determine their value in classifying the disease and to monitor (residual) leukemic cells [41,42]. The evaluation of (leukemia-specific) immunoreactive cells has been shown to further contribute to evaluating therapy efficiency, to quantifying antileukemic reactions and to improving therapy options and prognoses [43,44].

### 3.1. DC-Based Therapies as Promising Therapy Options

Due to their ability to mediate innate and the adaptive immune response, DCs’ and DC_leu_s’ therapeutic potential has been widely recognized [5,6,7,9]; DC vaccinations with monocyte-derived DCs, as well as DC_leu_ converted from patients’ myeloid blasts, have shown immunological effects in vivo [9,10,45,46].

### 3.2. Improved Activation of the Adaptive and Innate Immune System with Kit-Treated WB

#### 3.2.1. Ex Vivo DC Generation and (Antileukemic) Immune Cell Activation

DC (subtypes) from healthy as well as patients’ WB samples can be generated in the presence of Kits (compared to controls) [10,27,47,48] (Figure 1). Combinations of GM-CSF and PGE1 (Kit-M) or GM-CSF and Picibanil (Kit I), added to healthy/leukemic whole blood, use the soluble microenvironment in WB (containing, e.g., cytokines and chemokines) as an additional source for response modifiers to generate DC/DC_leu_. Both Kits provide danger signaling combined with maturation signaling, which guide cells’ differentiation towards DC or DC_leu_, respectively, as published before [10,48]. The detailed functional pathways of (PGE1-/Picibanil-containing) DC-generating methods are not known. The induction of monocyte or blast proliferation was not seen, thereby indicating that Kits do not induce blast or monocyte proliferation; however, they give rise to significantly increased frequencies of leukemia- or monocyte-derived, mature DC, as already shown [10,11] (Figure 1). According to our findings (also confirmed here), we can deduce that Kit-M might trigger, with higher efficacy, improved/mediated antileukemic reactions compared to Kit I. Moreover, we show that the achieved antileukemic activities (achieved using patients’ samples at first diagnosis) are independent of patients’ sex, cytogenetic risk and blast counts (patients with leukemia in remission might profit from these Kit-mediated effects; (residual) blasts are converted to DC_leu_ and trigger the immune system specifically against blasts). This has to be proven in a clinical trial [17,21].

Furthermore, we confirm that adaptive and innate immune cells from healthy and patients’ samples pretreated (vs. not pretreated) with Kits were regularly (significantly) activated after MLC, giving rise to (significantly) increased frequencies of activated cells of the innate and adaptive immune system in AML samples, pointing to Kit-mediated activation of immune cells and the generation of memory cells (Figure 2), as shown before [9,10,11,21,27,30,48]. We also found higher frequencies of activated immune-reactive cells in healthy (compared to patients’) samples, as already shown [27], possibly pointing to immunological activation against various bacterial, viral or mycotic targets [49,50] (Figure 2). Compared to uncultured cells, activation of immune cell subtypes after MLC was seen, due to the influence of IL-2, as expected [51] (Figure 2 and Figure 3).

#### 3.2.2. Higher β7 Expression in Immune Cells after MLC with Kit-I and Kit-M Treatment of Healthy and Patients’ WB Samples

β7 expression in uncultured T-cells has already been shown to correlate with cell cytotoxicity against leukemic (blasts) and other (intestinal/intraepithelial cells) targets [15,38,39,40]. We found that healthy donors’ samples showed significantly higher frequencies of β7 in T-cells compared to patients’ samples before cultivation, as well as in cells after culture (control, Kit-M) (Figure 3); this confirms the findings of Vogt et al. obtained before/after MLC with blast-containing MNC pretreated using DC-generating methods (MCM-Mimic, Picibanil and Ca-Ionophore) as stimulator cells [15]. These increased frequencies of β7 expression in healthy cells compared to those of patients might indicate a detrimental effect of leukemic immunosuppression on β7 expression in immune cells [49]. Moreover, our data show (significantly) increased frequencies of β7-expressing CD3^+^, T_non-naive_ and T_cm_ in healthy and patients’ samples after MLC with Kit-M- and Kit-I-pretreated (vs. control) WB (Figure 3). These data might indicate the involvement of β7-expressing immune cells in immune functionality in healthy as well as patients’ samples (Figure 3).

Comparing the DC-generating potential of Kit-I and Kit-M, we found higher frequencies of DC after Kit-I treatment in healthy samples and lower frequencies of DC_leu_ in patients’ samples. This might point to lower efficiency of Kit-I in generating functional DC (Figure 1). These results confirm the unpublished data of Ugur et al. [17], who also showed lower functionality compared to Kit-M.

#### 3.2.3. Increased Production of (Antigen-Specific) Degranulating or Intracellular Cytokine-Producing Immune Cells after MLC of Kit-M-Pretreated Healthy and Patients’ WB Samples

According to their biological function, DCs can help to overcome the anergy of immune-reactive cells and prime effector cells against their targets. Efficacy can (and has to) be demonstrated by induced/increased immune-reactive cells and decreased blast counts. (Functional) specific effects, mediated by DC/DC_leu_ have to be quantified (compared to controls). In the case of DC (loaded with tumor antigens) or DC_leu_, their capability to activate the immune system specifically against leukemic cells has to be evaluated after (T-cell-enriched) mixed lymphocyte culture (MLC) using DC/DC_leu_ as stimulator cells. The assays to detect leukemia-specific activations are cytokine secretion assays, degranulation assays, intracellular cytokine assays, ELISPOT, TETRAMER, etc. [52]. Adding leukemia-associated antigens (e.g., WT1 and PRAME) to cultures with/without the cultivation of cells can help to detect/enrich low frequencies of specific cells [21,30,53].

DEG und INTCYT assays are useful for demonstrating the antigen-specific activation of immunoreactive cells by measuring cell degranulation and intracellular cytokine production (with and without stimulation with LAA) [21,30]. We already showed significant activation of antigen-specific degranulating or intracellularly cytokine-producing innate and adaptive immune cells after MLC with Kit-M-pretreated (vs. untreated) WB (and low frequencies in uncultured cells) [21,30]. Studying β7-expressing immune cells, we observed higher frequencies of degranulating β7-expressing T- and NK-cells, CIK-cells and intracellularly cytokine-producing T-cells (in healthy and/or patients’ samples) after MLC(M) (compared to the control) (Figure 4A,B), with higher frequencies in healthy compared to patients’ samples (Figure 4A,B). Together with the findings after the Kit-M-mediated MLC of increased β7 expression, this might indicate (specific) involvement of these cells in immune reactivation, in healthy as well as in AML patients.

While the addition of LAA (to patients’ samples) or SEB (to healthy samples) antigen stimulation [21,29] led to higher frequencies of antigen-specific cells, especially in uncultured cell samples, it did not significantly enhance the Kit-M-mediated effects after MLC with respect to induced antigen-specific cells compared to the control (data not shown). This confirms the previous data, which show that Kit-M pretreatment of WB (going along with the generation of DC) stimulates and activates immune-reactive cells and compensates for LAA- or SEB-triggered activating effects [21].

In principle, the induction of leukemia-specific cells can be also detected in vivo by applying the methods given above [54]. Several groups have shown that AML patients treated with DC/DC_leu_ showed significant induction of (leukemia-specific) cellular and humoral immunity, going along with reduced blasts and prolonged overall survival (e.g., [54,55,56,57,58,59], all cited in [52]). Depending on the methods used, significant induction of the leukemia-specific immune response was defined as a 25 (50)% increase in the frequency of leukemia-specific cells compared to the initial count, or significantly higher frequencies of induced specific cells compared to untreated controls (e.g., [55,60,61]).

#### 3.2.4. Increased Blastolytic Functionality of Immune-Reactive Cells in Kit-Pretreated Samples after MLC

The hardest proof of induced or improved antileukemic activity (compared to controls) is the detection of improved blast lysis compared to the controls (e.g., after MLC). This can be achieved via chrome release, fluorolysis or other assays [62]. The Kit pretreatment of blasts in WB has already been shown to improve the antileukemic activation of immunoreactive cells after MLC, going along with significantly increased blast lysis compared to the controls [11,17,21,30,48].

Here, we found (nearly significantly) more cases with a superior blastolytic effect of immune-reactive cells after MLC(M) compared to the control (*p* = 0.056). Kit-I pretreatment was less effective compared to Kit-M pretreatment in giving rise to improved blastolytic functionality of the immune-reactive cells (Figure 5).

Blast lysis was, for some cases, superior after 3 h of incubation of target with effector cells, and for other cases, after 24 h. These effects could be attributed to different, independent blastolytic mechanisms, namely the faster perforin/granzyme pathways (leading to blast lysis predominantly after 3 h of coincubation of target with effector cells) and the slower Fas/FasL pathways (leading to blast lysis predominantly after 24 h of coincubation of target with effector cells) [17,21,63].

Summarizing the blastolytic effects achieved (after 3 h or 24 h of coincubation of effector with target cells), our data support previous data, which show that the Kit-M pretreatment of WB improves the blastolytic activity of immunoreactive cells in cell cultures [17,21].

### 3.3. Potential of β7 Monitoring

#### 3.3.1. β7 Expression in Immune-Reactive Cells as a Clue to Higher Susceptibility to Chemotherapy and Kit Treatment

Compared to patients with AML/MDS, the frequencies of β7-expressing cells were higher in healthy donors with a healthy immune system, unaffected by the immunosuppressive effect of neoplastic cells [50,64]. Generally, although the differences were not significant due to low case numbers, AML patients at first diagnosis who achieved (vs. those who did not achieve) leukemia remission after induction chemotherapy presented with higher frequencies of β7-expressing immune cell subtypes in uncultured WB samples; moreover, patients’ assignment to the favorable vs. adverse cytogenetic risk group went along with higher frequencies of β7^+^ T_cm_ in uncultured patients’ WB (Figure 6). This could point to an influence of β7-expressing cells in contributing to a healthy functional immune system [15,38,40].

#### 3.3.2. β7 Expression as a Marker for Improved Blast Cytotoxicity

We were able to find positive correlations of achieved blast lysis with frequencies of β7-expressing cells after MLC(M). These results confirm previous findings that showed higher blastolytic cytotoxicity in samples with higher frequencies of β7 expression using different (Kit-independent) DC-generating methods, and could also point to involvement of β7 expression in the mediation of superior blastolytic functionality in immune cells [15]. In addition, the improved blast lysis correlated positively with the increased degranulation activity (Figure 7C), which further demonstrates the antileukemic potential of these Kit-induced β7-expressing immune cells and the enhancing effect of Kit-M pretreatment on antileukemic functionality. Similar correlations could also be found for cells after MLC (Ugur [17] and personal communication; publication in preparation).

Possible correlations of β7 expression with cytotoxicity have been discussed in the context of autoimmune processes [38,40]. In the case of malignant diseases such as AML, it was hypothesized that a correlation between β7 expression and cell toxicity might have potential as a marker indicating the antileukemic functionality of immune cells [15]. This hypothesis could be further supported by our finding that β7 expression also correlated with more pronounced antileukemic intracellular cytokine production of β7-expressing cells after MLC(M) (Figure 7B) (which also correlated with higher antileukemic functionality, as shown before [21]).

We can state that the Kit-M-induced activation of immune cells after T-cell-enriched MLC goes along with increased β7 expression in (leukemia-specific) immune cells and correlates with improved antileukemic activity.

## 4. Materials and Methods

### 4.1. Patients and Healthy Sample Acquisition

Heparinized peripheral WB samples (provided by the university hospitals of LMU in Munich, Augsburg, Oldenburg and Tuebingen, as well as the Rotkreuzklinikum in Munich) were taken from patients diagnosed with acute myeloid leukemia (AML) or myelodysplastic syndrome (MDS) and healthy volunteers. Patients’ consent was given according to Helsinki guidelines and the vote of the Ethics Committee of LMU in Munich (vote number: 33905).

We included 31 patients with AML or MDS in acute stages of disease and 18 healthy volunteers (as given in Table 2). Patient age at sample acquisition was, on average, 60.8 (29–98) years, and the age of healthy probands was 31.3 (20–56) years. The male:female ratio of patients was 1:0.8, and 1:1.25 in healthy individuals. The average peripheral blood (PB) blast count of patients was 33 (10–94) %Bla/cells. AML cases were classified according to the FAB classification system [65], and MDS cases according to the WHO classification system [66]. Assessments for AML patients were further risk-classified according to the ELN classification [67] in favorable, intermediate and adverse subgroups, and for MDS patients according to the IPSS-R classification system [5,68] (Table 2).

### 4.2. Initial Sample Preparation

Mononuclear cells (MNCs) [10,69] were isolated from WB according to standard-preparations. T-cells were prepared using MACS microbead technology (Miltenyi Biotec, Bergisch Gladbach, Germany) [70]. MNC and T-cells were frozen for later experiments, as described [10,69,70].

### 4.3. Cultivation of Dendritic Cells (DC) and Leukemia-Derived Dendritic Cells (DC_leu_)

DC and DC_leu_ were cultured with (vs. without) combinations of response modifiers (Kit-M and Kit-I), as described [10].

Kit-M consisted of GM-CSF (final concentration (fc): 800 U/mL, granulocyte macrophage colony stimulating factor, Sanofi-Aventis, Frankfurt, Germany) and PGE1 (fc: 1 µg/mL, prostaglandin E1, Santa Cruz Biotechnology, Dallas, Texas, USA), and Kit-I consisted of GM-CSF (fc: 800 U/mL) and Picibanil (fc: 10 µg/mL, Chugai Pharmaceutical Co., Kajiwara, Japan). Response modifiers were added on Day 0 and on Day 2–3 [27]. After 6–8 days, the cells were harvested, evaluated via flow cytometry and used for further experiments [10]. Cells will be referred to as DC(M) for Kit-M-treated WB, DC(I) for Kit-I-treated WB and DC(C) for the control with untreated WB.

### 4.4. Cultivation of Cells in Mixed Lymphocyte Cultures (MLC)

After seven days of DC generation (Kit-treated or untreated WB) the cultivation of MLC was started with previously frozen/thawed T-cells in the presence of 50 U/mL IL-2, as described [21,30].

After 6–8 days, cells were harvested, immune cell subtypes were quantified, and degranulation (DEG), intracellular cytokine (INTCYT) and cytotoxicity fluorolysis assays (CTX) were performed, as described [10,27]. Cells before MLC will be referred to as MLC(UC) and those after MLC as MLC(CC) for the control with untreated WB. MLC(M) will refer to MLC with Kit-M-pretreated WB, and MLC(I) to MLC with Kit-I-pretreated WB.

### 4.5. Degranulation Assay (DEG) and Intracellular Cytokine Assay (INTCYT)

Cells (uncultured WB or T-cell-enriched MLCs, stimulated with or without Kit-treated WB after MLC) were mixed and incubated in parallel with those without leukemia-associated antigens (LAA) (PepTivator WT1 (fc: 0.6 nmol/mL, Miltenyi Biotec, Bergisch Gladbach, Germany) and PepTivator PRAME (fc: 0.6 nmol/mL, Miltenyi Biotec, Bergisch Gladbach, Germany)) for patients’ samples and with staphylococcal enterotoxin B (SEB, fc: 10 µg/mL, Sigma-Aldrich, St. Louis, Missouri, USA), as described [21,30,71].

An FITC-conjugated antibody against CD107a (BioLegend, San Diego, CA, USA) was used to detect cell degranulation as a marker for cell cytotoxicity, as described [19,72]. Finally, cells were harvested and analyzed via flow cytometry [30].

To quantify intracellular cytokine production, antibodies against tumor necrosis factor alpha (TNFα) [20] and interferon gamma (IFNγ) [21] were used: PE/Cy 7-TNFα antibodies (BioLegend, San Diego, CA, USA) and PE-IFNγ antibodies (BioLegend, San Diego, CA, USA)). To stop the cellular cytokine secretion, after one hour of incubation time, Brefeldin A solution (fc: 5 µg/mL, BioLegend, San Diego, CA, USA) was added according to the manufacturer’s instructions. Finally, cells were harvested and analyzed via flow cytometry [21].

### 4.6. Cytotoxicity Fluorolysis Assay (CTX)

CTX was performed to test the ability of effector cells (T-cell-enriched cells, stimulated with Kit-M-treated or untreated WB) to lyse blast target cells.

Kit-M-treated (vs. untreated as a control) patients’ WB (as ‘stimulator cells‘) and patients’ T-cells (frozen at the start of the experiments) were used as the main ‘effector cells‘ to be activated by MLC. AFTER MLC functionality was measured: cells in MLC (after stimulation with Kit-M-treated (vs. untreated) WB cells) were ‘effector cells‘. Blast-containing MNC (frozen at the start of the experiments) were added as ‘target cells‘. Blasts were stained using patient-related blast markers and were quantified via flowcytometry. Finally, the frequencies of viable (7AAD-negative) blast cells after the incubation of ‘target cells’ (blast-containing MNC) with ‘effector cells’ (T-cell-enriched MLC stimulated (vs. not stimulated) with Kit-M before were quantified, and blast lysis calculated according to previous publications, e.g., [21].

Finally, the difference in viable blast-cells in the main and the control samples was defined as ‘blast lysis’. The ‘lysis improvement’ was determined by comparing the achieved ‘blast lysis’ after MLC with, compared to without, Kit-pretreated WB [10,17,32].

### 4.7. Quantification of Cells Using Flow Cytometry

Before or after culture, cells were stained using fluorochrome-labeled monoclonal antibodies and quantified using a FACSCalibur four-channel flow cytometer as described [12,16]. The antibodies were conjugated with FITC (fluorescein-isothiocyanate), PE (phycoerythrin), PE/Cy 7 (phycoerythrin/cyanine 7) and APC (allophycocyanin). The FITC-labeled antibodies were: IgG*, CD34*, CD65*, CD33*, CD117***, CD15*, CD56***, CD3**, CD71*, ipo38****, CD19*, CD107a***, CD4**, CD45RO* and CD14*. The PE-labeled antibodies were: IgG*, CD 117*, CD80*, CD83*, CD56*, CD206*, CD3*, IFNγ*** and CD4**. The PE/Cy 7-labeled antibodies were: IgG*, CD15**, CD117*, CD19*, CD34*, CD197**, CD56*, CD4*, TNFα***, CD3* and CD14**. The APC-labeled antibodies were: IgG*, CD206**, CD80***, CD209**, CD83**, CD34*, CD117*, CD14*, CD56*, CD69**, β7**, CD45RO***, CD4**, CD3* and CD19*. The antibodies were supplied by Beckman Coulter (*, Brea, CA, US), BD Biosciences (**, San Jose, CA, US), BioLegend (***, San Diego, CA, US) and Santa Cruz Biotechnology (****, Dallas, TX, US). For the detection of viable cells, 7AAD** was used.

For every patient, highly expressed blast markers (e.g., CD34, CD117, CD65, CD56 or CD15) and DC markers with low/no expression of blasts (e.g., CD80, CD83, CD206 or CD209) were selected to quantify DC or DC_leu_ subpopulations after culture [12,16].

For the detection of intracellular markers (ipo38 and INTCYT), the FIX & PERM Cell Fixation and Cell Permeabilization Kit (Thermo Fisher Scientific, Darmstadt, Germany) was used. Isotype samples served as controls [10,73].

### 4.8. Statistical Analysis

The analysis of flow cytometric data was conducted using BD CellQuest Pro software (Becton Dickinson, Heidelberg, Germany). Statistical analyses, including the calculation of means, standard deviations and significance, were conducted using Excel 2010 (Microsoft, Redmond, Washington, USA) and SPSS Statistics 26 (IBM, Armonk, New York, NY, USA). Differences and correlations between groups were analyzed using the paired-*t*-test and the Wilcoxon–Mann–Whitney U test. Correlation analyses were conducted using Pearson correlation and Spearman correlation. Highly significant differences were defined in cases with *p*-values ≤ 0.005 and significantly different cases with *p*-values between 0.05 and 0.0051.

## 5. Conclusions

β7, as a subunit of the integrin receptor, is expressed in several subtypes of healthy and patients’ cells in the adaptive and innate immune system. Pretreatment of AML/MDS patients’ WB samples with blast-modulating Kits (vs. no Kits) increased the frequencies of DC/DC_leu_, which ultimately increased the frequencies of (leukemia-specific degranulating or cytokine-producing) β7-expressing T- or NK/CIK-cell subtypes after T-cell-enriched MLC. The frequencies of the generated/activated β7-expressing cells correlated ex vivo with the provision of leukemia-specific/antileukemic cells after (T-cell-enriched) MLC with Kit-pretreated (vs. untreated) WB, and in vivo with achieved (vs. not achieved) remission after induction chemotherapy, and with patients’ allocations to favorable (vs. unfavorable) risk types. (Due to the low case numbers available, not all results showed significant results; however, there were always clear differences between groups.)

## Figures and Tables

**Figure 1 ijms-24-00463-f001:**
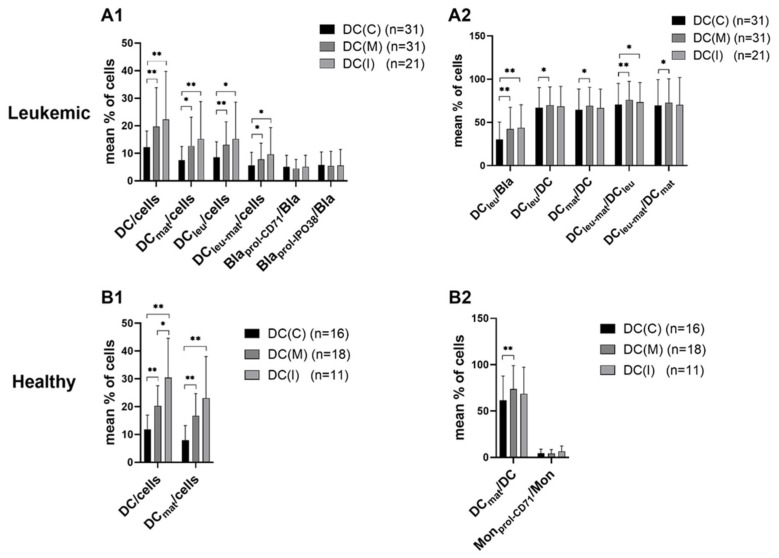
Generation of DC/DC_leu_ from (**A1**) leukemic and (**B1**) healthy WB with and without Kits. WB samples were cultured for 7 days with Kit-M or Kit-I or without added Kits as control. Results with Kit-M (DC(M)) or Kit-I (DC(I)) or without added Kits as control (DC(C)) are given. Mean frequencies ± standard deviation of DC subtypes in (**A2**) leukemic (AML/MDS) and (**B2**) healthy samples are given; n—number of cases. Differences were considered as significant (*) when *p* ≤ 0.05 and as highly significant (**) when *p* ≤ 0.005. Abbreviations of cell subpopulations are given in Table 1.

**Figure 2 ijms-24-00463-f002:**
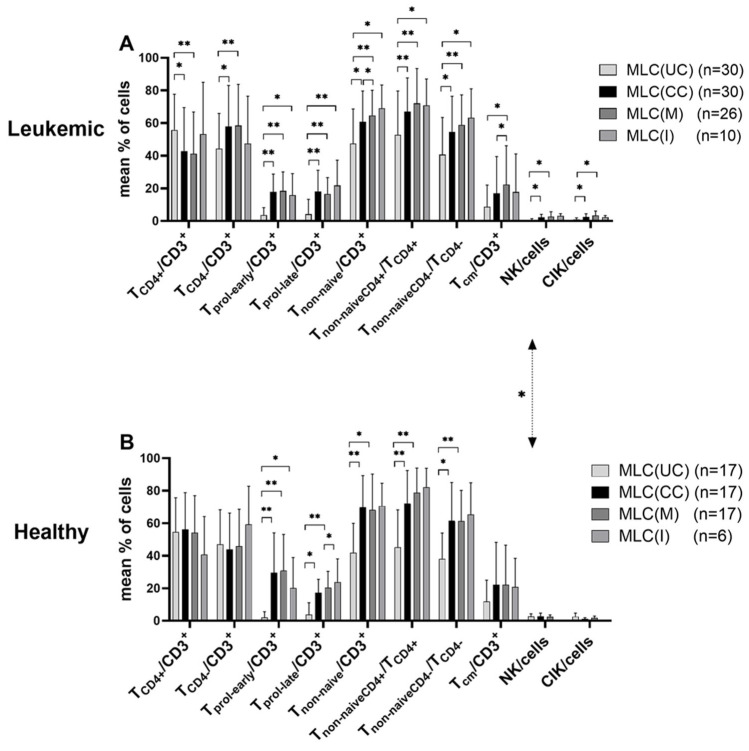
Composition of immune-reactive cells before and after T-cell-enriched MLC using (**A**) leukemic and (**B**) healthy WB with or without Kit pretreatment. Cells were analyzed via flow cytometry before and after 7 days of (T-cell-enriched) MLC with Kit-pretreated or untreated WB and IL-2. Cells before MLC from WB without added Kits as control (MLC(UC)), and cells after MLC, from WB pretreated with Kit-M (MLC(M)), Kit-I (MLC(I)) or without added Kits as control (MLC(CC)), are given. Mean frequencies ± standard deviation of immune-reactive cell subpopulations in (**A**) leukemic (AML/MDS) and (**B**) healthy samples are given; n—number of cases. Differences were considered as significant (*) when *p* ≤ 0.05 and as highly significant (**) when *p* ≤ 0.005. Double-sided arrows give (significant) differences between defined healthy and leukemic immune-reactive cell subtypes. Abbreviations of cell subpopulations are given in Table 1.

**Figure 3 ijms-24-00463-f003:**
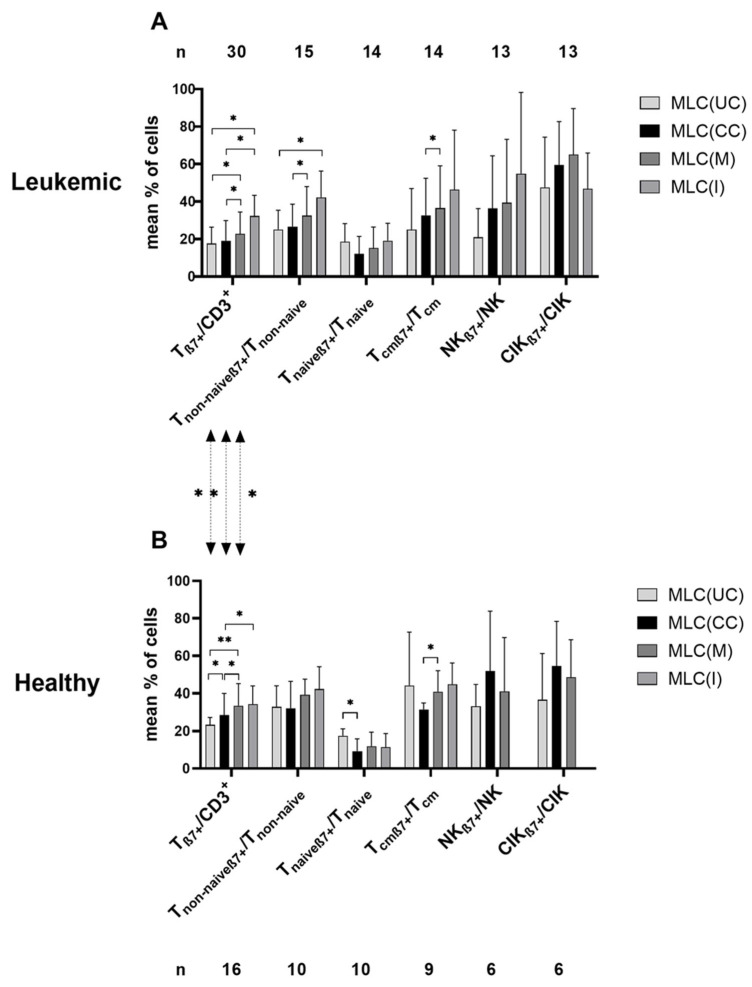
Composition of β7-expressing immune-reactive cells before and after T-cell-enriched MLC using (**A**) leukemic and (**B**) healthy WB with and without Kit pretreatment as stimulator cells. Cells were analyzed via flow cytometry before and after 7 days of (T-cell-enriched) MLC with Kit-pretreated or untreated WB and IL-2. Cells before MLC from WB without added Kits as control (MLC(UC)), and cells after MLC, from WB pretreated with Kit-M (MLC(M)), Kit-I (MLC(I)) or without added Kits as control (MLC(CC)), are given. Mean frequencies ± standard deviation of immune-reactive cell subpopulations in (**A**) leukemic (AML/MDS) and (**B**) healthy samples are given; n—number of cases. Differences were considered as significant (*) when *p* ≤ 0.05 and as highly significant (**) when *p* ≤ 0.005. Double-sided arrows give (significant) differences between defined healthy and leukemic immune-reactive cell subtypes. Abbreviations of cell subpopulations are given in Table 1.

**Figure 4 ijms-24-00463-f004:**
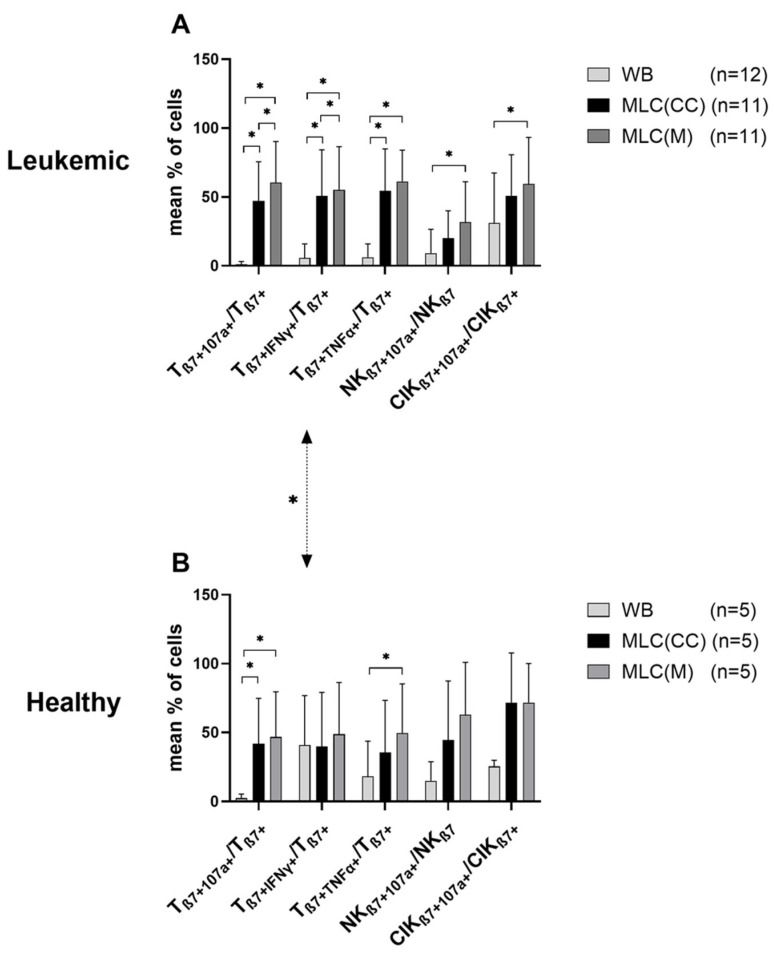
Composition of β7-expressing degranulating or intracellular cytokine-producing immune-reactive cells in uncultivated WB and after T-cell-enriched MLC using (**A**) leukemic and (**B**) healthy WB with or without Kit pretreatment as stimulator cells. Degranulation and intracellular cytokine production were quantified via flow cytometry in untreated and uncultivated WB as well as after 7 days of (T-cell-enriched) MLC with Kit-pretreated or untreated WB and IL-2. Results without LAA or SEB stimulation are given. Uncultivated cells in WB and cultivated cells after MLC from WB pretreated with Kit-M (MLC(M)) or without added Kits as control (MLC(CC)) are given. Mean frequencies ± standard deviation of immune-reactive cell subpopulations in (**A**) leukemic (AML/MDS) and (**B**) healthy samples; n—number of cases. Differences were considered as significant (*) when *p* ≤ 0.05. Double-sided arrows give (significant) differences between defined healthy and leukemic immune-reactive cell subtypes. Abbreviations of cell subpopulations are given in Table 1.

**Figure 5 ijms-24-00463-f005:**
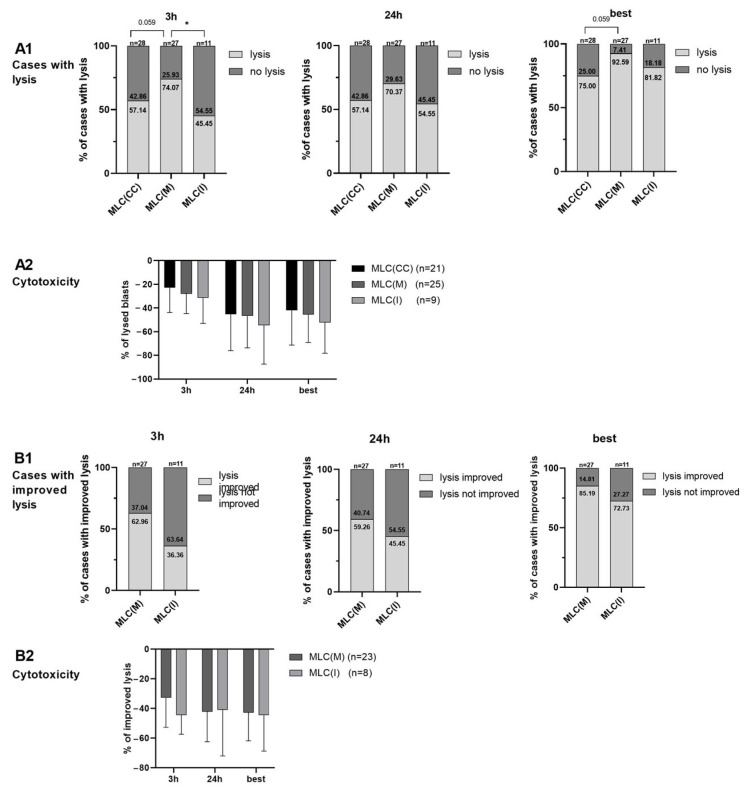
Blastolytic potential of immune-reactive cells after T-cell-enriched MLC using leukemic WB with and without Kit pretreatment as stimulator cells. For the cytotoxicity assay, target and effector cells were coincubated for a total of 24 h. Results after 3 h and 24 h and the ‘best of’ values after coincubation are given. Results after MLC from WB pretreated with Kit-M (MLC(M)) or Kit-I (MLC(I)) or without added Kits as control (MLC(CC)) are given. Percentages of cases (**A1**) with achieved (vs. non-achieved) blast lysis and (**B1**) with improved (vs. non-improved) blast lysis are given. Mean frequencies ± standard deviation of (**A2**) lysed blasts (in cases with lysis) and (**B2**) lysis improvement (in cases with improved lysis) are given; n—number of cases. Differences were considered as significant (*) when *p* ≤ 0.05. Abbreviations of cell subpopulations are given in Table 1.

**Figure 6 ijms-24-00463-f006:**
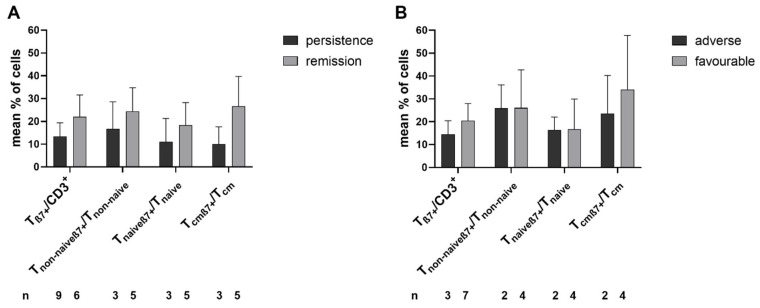
Composition of uncultivated β7-expressing immune-reactive cells in AML patients’ samples with patients subdivided into different groups at first diagnosis. Uncultured cells (MLC(UC)) were analyzed via flow cytometry. Mean frequencies ± standard deviation of β7-expressing immune-reactive cell subpopulations in patients with AML at first diagnosis with respect to patients’ (**A**) responses to chemotherapy and (**B**) allocation to cytogenetic ELN risk groups are given; n—number of cases. Abbreviations of cell subpopulations are given in Table 1.

**Figure 7 ijms-24-00463-f007:**
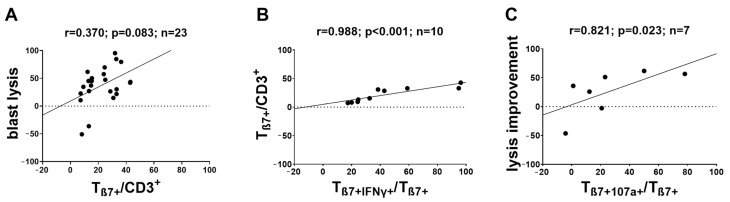
Correlations of antileukemic functionality with frequencies of (leukemia-specific) β7-expressing immune-reactive cells before and after T-cell-enriched MLC using Kit-M-pretreated (vs. untreated as control) leukemic WB as stimulator cells. Cells were analyzed via flow cytometry before and after 7 days of (T-cell-enriched) MLC with Kit-pretreated or untreated WB and IL-2. Results of uncultured cells before MLC (MLC(UC)), and cells after MLC, from WB pretreated with Kit-M (MLC(M)) or without added Kits as control (MLC(CC)), are given. Lysis (improvement) is given as the best of value after MLC(M) (compared to MLC(CC)). (**A**) Clear positive correlation of frequencies of T_β7+_/CD3^+^ with best blast lysis after MLC (with Kit-M-pretreated leukemic WB). (**B**) Significant positive correlation of frequencies of T_β7+_/CD3^+^) with T_β7+IFNγ+/_T_β7+_ after MLC (with Kit-M-pretreated leukemic WB. (**C**) Significant positive correlation of frequencies of T_β7+107a+/_T_β7_ with improved blast lysis after MLC (with Kit-M-pretreated leukemic WB). r—correlation coefficient, p—significance, n—number of cases. Differences were considered as significant when *p* ≤ 0.05 and as highly significant when *p* ≤ 0.005. Abbreviations of cell subpopulations are given in Table 1.

**Table 1 ijms-24-00463-t001:** Cell subpopulations.

Group	Subgroup	Acronym	Markers	Abbrev. (Referred to Cell Fraction)	Reference
**T-cells**	T-cells	T	CD3^+^	CD3^+^cells	[14]
	Transferrin-R-positive late-proliferating T-cells	T_prol-late_	CD3^+^CD71^+^	T_prol-late_/CD3^+^	[14]
	Type II C- type lectin-positive early-proliferating T-cells	T_prol-early_	CD3^+^CD69^+^	T_prol-early_/CD3^+^	[14]
	CD4-positive T-cells	T_CD4+_	CD4^+^CD3^+^	T_CD4+_/CD3^+^	[14]
	CD4-negative T-cells	T_CD4−_	CD4^−^CD3^+^	T_CD4−_/CD3^+^	[14]
	Non-naive T-cells	T_non-naive_	CD3^+^CD45RO^+^	T_non-naive_/CD3^+^	[15]
	Naive T-cells	T_naive_	CD3^+^CD45RO^−^	T_naive_/CD3^+^	[15]
	Central memory T-cells	T_cm_	CD3^+^CD45RO^+^CCR7^+^	T_cm_/CD3^+^	[15]
	CD4-positive non-naive T-cells	T_non-naiveCD4+_	CD3^+^CD4^+^CD45RO^+^	T_non-naiveCD4+_/T_CD4+_	[10]
	CD4-negative non-naive T-cells	T_non-naiveCD4−_	CD3^+^CD4^−^CD45RO^+^	T_non-naiveCD4−_/T_CD4−_	[10]
**blasts**	Blasts	Bla	e.g: CD15, CD34, CD65, CD117,	Bla/cells	[16]
	Proliferating blasts	Bla_prol-CD71_	CD71^+^Bla^+^DC^−^	Bla_prol-CDl71_/Bla	[17]
	Proliferating blasts	Bla_prol_^−^_IPO38_	IPO38^+^Bla^+^DC^−^	Bla_prol-IPO38_/Bla	[17]
**dendritic cells**	Dendritic cells	DC	CD80, CD206, CD209, CD83	DC/cells	[12]
	Mature DC	DC_mat_	DC^+^CCR7^+^	DC_mat_/cells DC_mat_/DC	[13]
	Leukemia-derived DC	DC_leu_	Bla^+^DC^+^	DC_leu_/cells DC_leu_/DC DC_leu_/Bla	[12]
	Mature leukemia-derived DC	DC_leu-mat_	Bla^+^DC^+^CCR7^+^	DC_leu-mat_/cells DC_leu-mat_/DC_leu_ DC_leu-mat_/DC_mat_	[17]
**others**	Cytokine-induced killer cells	CIK	CD3^+^CD56^+^	CIK/cells	[10]
	Natural killer cells	NK	CD3^−^CD56^+^	NK/cells	[10]
	Monocytes	Mon	CD14^+^	Mon/cells	[10]
	Proliferating monocytes	Mon_prol-CD71_	CD71^+^CD14^+^DC^−^	Mon_prol-CD71_/Mon	
	B-cells	B	CD19^+^	CD19^+^/cells	[10]
**Integrin beta 7 (β7)**	β7-positive T-cells	T_β7+_	β7^+^CD3^+^	T_β7+_/CD3^+^	[15]
	β7-positive CIK-cells	CIK_β7+_	β7^+^CD3^+^CD56^+^	CIK_β7+_/CIK	
	β7-positive NK-cells	NK_β7+_	β7^+^CD3^−^CD56^+^	NK_β7+_/NK	[18]
	β7-positive non naive T-cells	T_non-naiveβ7+_	β7^+^CD3^+^CD45RO^−^	T_non-naiveβ7+_/T_non-naive_	[15]
	β7-positive naive T-cells	T_naiveβ7+_	β7^+^CD3^+^CD45RO^+^	T_naiveβ7+_/T_naive_	[15]
	β7-positive central memory T-cells	T_cmβ7+_	β7^+^CCR7^+^CD3^+^CD45RO^+^	T_cmβ7+_/T_cm_	[15]
**DEG**	CD107a-positive β7-positive T-cells	T_β7+107a+_	CD107a^+^β7^+^CD3^+^	T_β7+107a+_/T _β7+_	[19]
	CD107a-positive β7-positive CIK-cells	CIK_β7+107a+_	CD107a^+^β7^+^CD3^+^CD56^+^	CIK_β7+107a+_/CIK_β7+_	[19]
	CD 107a-positive β7-positive NK-cells	NK_β7+107a+_	CD107a^+^β7^+^CD3^−^CD56^+^	NK_β7+107a+_/NK_β7+_	[19]
**INTCYT**	TNF alpha-positive β7-positive T-cells	T_β7+TNFα+_	TNFa^+^β7^+^CD3^+^	T_β7+TNFα+_/T_β7+_	[20]
	IFN gamma-positive β7-positive T-cells	T_β7+IFNγ+_	IFNg^+^β7^+^CD3^+^	T_β7+IFNγ+_/T_β7+_	[21]

Abbrev. cell subpopulations; marker combinations (measured by FACS) defining the different cell subpopulations.

**Table 2 ijms-24-00463-t002:** Characteristics of patients and healthy individuals.

Diagn.	No.	Age	Sex	Subtype	Stage	Ic bla	Blast Phenotype [cd]	Risk Class.	Response	Exp.
**AML**	1444	35	f	p/M1	dgn.	41	33, 65, **15**, **34**, 117	favorable	yes	DC; MLC; CTX
**AML**	1540	83	f	p/M1	dgn.	32	13, 34, 33, 15, **117**, **56**	favorable	no	DC; MLC; CTX; D/I
**AML**	1427	52	m	p/M2	dgn.	94	13, 33, 34, **117**	favorable	no	DC; MLC; CTX
**AML**	1541	82	f	p/M2	dgn.	15	15, **34**, **117**	interm.	no	DC; MLC; CTX; D/I
**AML**	1442	73	f	p/M4	dgn.	15	33, 13, 34, **117**, **15**	interm.	yes	DC; MLC; CTX
**AML**	1459	54	m	p/M4	dgn.	51	33, 64, **15**, 4, 56, 14	favorable	yes	DC; MLC; CTX
**AML**	1430	79	m	p/M5	dgn.	62	13, 33, **34**, **117**	favorable	nd	DC; MLC; CTX
**AML**	1432	34	m	p/M5	dgn.	57	**34**, 13, 33, 64, 4	interm.	yes	DC; MLC; CTX
**AML**	1466	47	f	p/M5	dgn.	11	33, 15, 13, **117**, **34**	adverse	yes	DC; MLC; CTX
**AML**	1575	62	f	p/M5a	dgn.	75	14, **56**, 64, **65**, 4	interm.	yes	DC; MLC; CTX; D/I
**AML**	1452	44	m	p/nd	dgn.	11	**34**, **117**, 33, 13	interm.	no	DC; MLC; CTX
**AML**	1568	29	m	p/nd	dgn.	69	**34**, 117, 33, 13, 19, 20, **65**	interm.	no	DC; MLC; CTX; D/I
**AML**	1570	36	f	p/nd	dgn.	11	**34**, **117**, **65**, 13, 33	favorable	no	DC; MLC; CTX; D/I
**AML**	1492	52	f	s/M2	dgn.	38	117, **34**, 13, 33, 7, **15**	nd	no	DC; MLC; CTX
**AML**	1542	58	f	s/M4	dgn.	52	13, 33, **34**, 117, 15, 65, 64, 2, **56**, 14	adverse	no	DC; MLC; CTX; D/I
**AML**	1426	61	f	s/M5	dgn.	34	13, 33, **34**, 64, **117**, 14	adverse	yes	DC; MLC; CTX
**AML**	1464	72	m	s/nd	dgn.	38	**34**, 117, 13	nd	nd	DC; MLC; CTX
**AML**	1555	46	f	s/nd	dgn.	20	33, 14, **15**, **117**, 13	favorable	nd	DC; MLC; D/I
**AML**	1574	56	m	s/nd	dgn.	41	**34**, **117**, 15, 19	nd	no	DC; MLC; CTX; D/I
**AML**	1571	61	m	p/M2	rel.	18	**117**, 33, 13, 7	nd	yes	DC; MLC; CTX; D/I
**AML**	1424	37	f	p/M4	rel.	13	13, 14, 33, **117**	nd	nd	DC; MLC; CTX
**AML**	1548	87	m	p/M5a	rel.	12	33, **15**, 117, 34, **56**	nd	no	DC; MLC; CTX; D/I
**AML**	1449	78	m	s/nd	rel.	32	65, 14, **15**, 33, 56, **34**	nd	nd	DC; MLC; CTX
**AML**	1482	75	m	s/nd	rel.	12	117, 13, 64, **15**, **117**, 33	nd	nd	DC; MLC; CTX
**AML**	1546	80	m	p/nd	pers.	22	33, **34**, 13, 117, 14, **65**	nd	no	DC; MLC;
**AML**	1470	67	m	p/nd	PR a. SCT	38	33, 117, **34**, 56, **65**	nd	no	DC; MLC
**AML**	1457	63	m	s/nd	rel. a. SCT	37	34, 117, 13, **65**, **15**	nd	no	DC; MLC; CTX
**AML**	1543	61	m	p/nd	rel. a. SCT.	38	13, 33, **117**, 56, **34**	nd	no	DC; MLC; CTX
**MDS**	1567	98	f	MDS	dgn.	14	34, 117, **15**, **65**, 56, 14	very high	nd	DC; MLC; CTX; D/I
**MDS**	1573	61	m	MDS	dgn.	12	**34**, **117**, 65, 13, 61	high	no	DC; MLC; CTX; D/I
**MDS**	1572	63	f	MDS-EB2	dgn.	10	**34**, **117**, 65, 33, 13	very high	no	DC; MLC; CTX; D/I
**healthy**	1417	34	f							DC; MLC
**healthy**	1418	22	m							DC; MLC
**healthy**	1421	27	f							DC; MLC
**healthy**	1422	20	f							DC; MLC
**healthy**	1425	27	m							DC; MLC
**healthy**	1428	56	f							DC; MLC
**healthy**	1429	22	f							DC; MLC
**healthy**	1431	22	m							DC; MLC
**healthy**	1436	25	m							DC; MLC
**healthy**	1440	20	f							DC; MLC
**healthy**	1448	27	f							DC; MLC
**healthy**	1458	21	f							DC; MLC
**healthy**	1544	22	m							DC; MLC; D/I
**healthy**	1545	32	m							DC; MLC; D/I
**healthy**	1547	46	f							DC; MLC; D/I
**healthy**	1566	54	f							DC; MLC; D/I
**healthy**	1576	55	m							DC; MLC; D/I
**healthy**	1578	32	m							DC; MLC

Diagn.—diagnosis; No.—sample number; m—male; f—female. Subtypes: p—primary AML; s—secondary AML. AML (FAB classification); MDS (WHO classification); rel. a. SCT—relapse after stem cell transplantation; rel.—relapse; dgn.—first diagnosis; pers.—persisting disease; PR—partial remission; nd—no data; Ic Bla—immunocytologically determined blasts; Risk Class: cytogenetic (AML ELN) and multifactorial (MDS IPSS-R) risk classification; Response—response to (induction) chemotherapy; Exp.—conducted experiments; MLC—mixed lymphocyte culture; DC—dendritic cell generation; CTX—cytotoxic fluorolysis assay; D/I—degranulation and intracellular cytokine assay.

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
