# Peer review of "Dendritic Cell-Triggered Immune Activation Goes along with Provision of (Leukemia-Specific) Integrin Beta 7-Expressing Immune Cells and Improved Antileukemic Processes"

_ijms, 2022, doi:10.3390/ijms24010463_

Round 1

Reviewer 1 Report

The manuscript by Rackl et al. addreses the possible use of DC-based therapies in leukemia. The authors generate leuDC and Mo-DC from whole blood cells of AML/MDC patients and healthy controls and use them as stimulators in MLC. Both types of DC induce T cell proliferation and blastolytic activation which correlated with higher frequencies of B7 expression, previously shown to play a role in cell cytotoxicity in malignant processes.  

The manuscript includes detailed sections on methodology and results. However, the length of the manuscript and the excessive focus on the descriptive aspects are weaknesses that need to be addressed. A shorter version, targeted on significance instead of description is recommended. The authors should address future approaches to validate a DC-based therapy beyond the in vitro experimental design.  

Author Response

Dear reviewer,

First, we thank you for the reviewing of our manuscript No. ijms-1825842.

I ‘General’. Please find attached all revisions (highlighted in yellow) including English language editing as well as shortening and reorganising of the manuscript, hereby focusing on relevant clinical data, as requested by reviewers. Please inform us in case the layout or formatting is lost (reviewer 3). With our statements below we address all points of criticism:

  1. Response to Reviewer 1:
  • General: “The manuscript by Rackl et al. addreses the possible use of DC-based therapies in leukemia. The authors generate leuDC and Mo-DC from whole blood cells of AML/MDS patients and healthy controls and use them as stimulators in MLC. Both types of DC induce T cell proliferation and blastolytic activation which correlated with higher frequencies of B7 expression, previously shown to play a role in cell cytotoxicity in malignant processes.”

e

  • Major comments:
  • The manuscript includes detailed sections on methodology and results. However, the length of the manuscript and the excessive focus on the descriptive aspects are weaknesses that need to be addressed. A shorter version, targeted on significance instead of description is recommended.”.

  • We have revised the whole manuscript and shortened whenever possible (see I).

  • The authors should address future approaches to validate a DC-based therapy beyond the in vitro experimental design”.

  • Thank you for this important comment: we are just in the process to plan a clinical trial for (therapy refractory) AML patients using our ‘blast modulating kit M’, composed of two clinically approved response modifiers. We added some sentences in the discussion.

  • Extensive editing of English language and style required”.

  • We have revised the whole manuscript regarding English language and style (see I).

We hope, our manuscript is ready to be published in your journal now

Kind regards,

Helga Schmetzer

Reviewer 2 Report

1. What is the functional difference between Kit (M), and Kit (I) ?

2. Which cell fraction in the WB is impacted the most by/most responsive to Kit(M)/KIT(I) treatment?

3. Among the patient samples, did blast percentage at the time of diagnosis significantly associate with response to KIT(M)/KIT(I) stimulation?

4. Please edit the style of Introduction, and Results to a more general format - grouping the Introduction into subsections, and bullet points, and including a "Prologue" in Results is non-canonical.

Author Response

Dear reviewer,

First, we thank you for the reviewing of our manuscript No. ijms-1825842.

I ‘General’. Please find attached all revisions (highlighted in yellow) including English language editing as well as shortening and reorganising of the manuscript, hereby focusing on relevant clinical data, as requested by reviewers. Please inform us in case the layout or formatting is lost (reviewer 3). With our statements below we address all points of criticism:

  1. Response to Reviewer 2:
  • Major comments:

  • : “What is the functional difference between Kit (M), and Kit (I)?”
  • Both Kit M and Kit I were deduced from well-known DC/DCleu generating protocols (see Schwepcke et al. 2022). They (both) use the soluble (cytokine/chemokine etc.) microenvironment of leukemic whole blood as source for DC-generating factors. In addition, GM-CSF, which is added to both kits, triggers myeloid (including blast) cells to differentiate healthy cells towards granulocytes/DC and blasts towards ‘leukemia-derived DC’. PGE1 (in Kit M) or Picibanil (in Kit I) both provide a ‘danger signaling’ combined with a ‘maturation’-signaling, that guide cells’ differentiation towards DC or DCleu as published before (e.g: Amberger 2019, Schwepcke 2022). The detailed functional pathways of PGE1 or Picibanil – containing DC methods or their derivative Kits are not known. According to our own new findings after MLC we might deduce, that Kit M might trigger with higher efficacy an antileukemic reaction compared to Kit I. This could also be confirmed here. We have included some sentences in the discussion.

  • : “Which cell fraction in the WB is impacted the most by/most responsive to Kit(M)/KIT(I) treatment?”
  • Kits (as derivates of DC/DCleu generating methods) trigger healthy stem cells and monocytes (= sources for DC) to differentiate to DC and blasts to differentiate to DCleu. (e.g: Schwepcke 2022)
  •  
  • : “Among the patient samples, did blast percentage at the time of diagnosis significantly associate with response to KIT(M)/KIT(I) stimulation?”
  • On average 50% of blasts are converted t DCleu in our ‘time window’. That means, we yield prove of concept that blasts are converted to DCleu, although not quantitatively. With respect to our aim (to treat patients after chemotherapy to convert residual blasts to DCleu, that trigger the immune system against (residual) blasts and to create a leukemic-specific memory, it is not important what happens at first diagnosis. However, we could not find correlations between blast count and blast lysis. What we have shown is that achieved antileukemic activity in settings at first diagnosis is independent of sex, age, cytogenetic risk or blast counts (Ugur et al. unpublished data).
  •  
  • : “Please edit the style of Introduction, and Results to a more general format - grouping the Introduction into subsections, and bullet points, and including a "Prologue" in Results is non-canonical”
  • We revised the whole manuscript, the style of the introduction and the results part (see also reviewer I) deleting many headings to subsections, bullet points, prologues.

  • : “Moderate English changes required”
  • We have revised the whole manuscript regarding English language and style (see also I).

We hope, our manuscript is ready to be published in your journal now

Kind regards,

Helga Schmetzer

Reviewer 3 Report

Elias Rackl et al. investigated the potential role of integrin beta-7 expression in antileukemic function, the story is interesting, and the data are abundant. Unfortunately, this reviewer has two major concerns about the manuscript: the manuscript was not organized well, and the data can not fully support the conclusion, for examples, if the differences are not significantly or the change fold is marginal, these data cannot convince this reviewer. Other minor issues are listed below:

1. Line 193, the “2” in CO2 and O2 should be subscript.

2. Please organize figures and figure legends.

3. For Figure 1A2, it is hard to observe the differentiations except the DC/Bla group. The differences were so marginal, did these differences make sense?

4. +and -should be super script in CD3+/TCD4-, for example.

5. How to calculate integrin beta-7 expression in Figure 3? Flow?

Author Response

Dear reviewer,

First, we thank you for the reviewing of our manuscript No. ijms-1825842.

I ‘General’. Please find attached all revisions (highlighted in yellow) including English language editing as well as shortening and reorganising of the manuscript, hereby focusing on relevant clinical data, as requested by reviewers. Please inform us in case the layout or formatting is lost (reviewer 3). With our statements below we address all points of criticism:

II: Response to Reviewer 3:

  • General: “Elias Rackl et al. investigated the potential role of integrin beta-7 expression in antileukemic function, the story is interesting, and the data are abundant.”

Major comments

  • : “Unfortunately, this reviewer has two major concerns about the manuscript: the manuscript was not organized well, and the data can not fully support the conclusion, for examples, if the differences are not significantly or the change fold is marginal, these data cannot convince this reviewer”
  • We have revised and reorganised the whole manuscript and also revised ‘statistically not significant data presentation’.

Minor comments

  • : “Line 193, the “2” in CO2 and O2 should be subscript.”
  • We have subscripted the “2”.

  • : “Please organize figures and figure legends”
  • We organized figures and figure legends.

3.: “For Figure 1A2, it is hard to observe the differentiations except the DC/Bla group. The differences were so marginal, did these differences make sense?”

  • We have checked all p-values again.

4.: “. “+” and “-” should be super script in CD3+/TCD4-, for example”

  • We have superscripted “+” and “-“.

5.: “How to calculate integrin beta-7 expression in Figure 3? Flow?”

  • We have added information about the calculation of integrin beta-7 expression in Figure 3: frequencies of ß7 positive cells in immune reactive cell (subsets) were calculated (e.g: for T cells: Tß7+/CD3+(=ß7 expressing cells in the CD3+ Tcell fraction). Moreover abbreviations are given in the text, explained and in addition presented in Table 3.

5.: “Moderate English changes required”

  • We have revised the whole manuscript regarding English language and style (see I).

We hope, our manuscript is ready to be published in your journal now

Kind regards,

Helga Schmetzer

Reviewer 4 Report

Among different strategies envisaged to treat cancers, including non-solid tumors like leukemia, immunotherapy, under different aspects, is one that lately has been regarded as quite promising. Within this scenario, dendritic cells (DCs) are extraordinarily important because, to some extent, they link the adaptive to the innate immunity. Basically, DCs capture, process, and present antigens to adaptive immune cells. Therefore, being able to handle the DCs and even more, loading them with cancer antigens, including leukemic ones, represents a potential weapon to counteract the leukemogenesis process. In their paper titled "Dendritic cell triggered immune activation goes along with provision of (leukemia-specific) Integrin beta 7 expressing immune cells and improved antileukemic processes" Rackl E. and colleagues generated DCs starting either from healthy blood monocytes, or leukemic blasts. Afterwards DCs were then used to stimulate T-cells and other immune cells. Eventually, the cytotoxicity of the latter against leukemic blasts was assessed. Overall, the manuscript presents several major flaws and it has been drafted in a way that looks a little bit awkward, at least the version that I have got to be downloaded. It looks like the layout/formatting has been lost, alternatively not properly formatted since the beginning. Shortly, very few examples: a) besides that legend of figures is poor and barely understandable. Moreover, the figures and their relative legends should pair, whereas in the manuscript this is not the case; b) line numbering too often interferes with the Y axis of the figures thus making it difficult to read what is written on the axis and thus interpret and read plots; c) it happens that in some figures (i.e. figure 2, figure 3) double-sided arrows appear but without any explanation in the legend. Quite odd? is it not?; d) Several oversights and typos are present throughout the main text.   The language needs profound and extensive revisions because the manuscript is quite difficult to be read fluently.   Statistics require to be carefully revised because it seems that blunders are present, due to some kind of discrepancies between the plots and asterisks. Eventually, I warmly advise the authors to avoid using "tendentially significant", because in statistics either there is a significant difference or not. Trends are too weak to support data and thus enabling the authors to draw solid and robust conclusions.

Author Response

Dear reviewer,

First, we thank you for the reviewing of our manuscript No. ijms-1825842.

I ‘General’. Please find attached all revisions (highlighted in yellow) including English language editing as well as shortening and reorganising of the manuscript, hereby focusing on relevant clinical data, as requested by reviewers. Please inform us in case the layout or formatting is lost (reviewer 3). With our statements below we address all points of criticism:

II: Response to Reviewer 4:

  • General: “Among different strategies envisaged to treat cancers, including non-solid tumors like leukemia, immunotherapy, under different aspects, is one that lately has been regarded as quite promising. Within this scenario, dendritic cells (DCs) are extraordinarily important because, to some extent, they link the adaptive to the innate immunity. Basically, DCs capture, process, and present antigens to adaptive immune cells. Therefore, being able to handle the DCs and even more, loading them with cancer antigens, including leukemic ones, represents a potential weapon to counteract the leukemogenesis process. In their paper titled "Dendritic cell triggered immune activation goes along with provision of (leukemia-specific) Integrin beta 7 expressing immune cells and improved antileukemic processes" Rackl E. and colleagues generated DCs starting either from healthy blood monocytes, or leukemic blasts. Afterwards DCs were then used to stimulate T-cells and other immune cells. Eventually, the cytotoxicity of the latter against leukemic blasts was assessed.”

Major comments

  • Overall, the manuscript presents several major flaws and it has been drafted in a way that looks a little bit awkward, at least the version that I have got to be downloaded. It looks like the layout/formatting has been lost, alternatively not properly formatted since the beginning.”
  • We apologize for the formatting and have revised the layout of the whole manuscript. Please inform us, in case the format is lost again in the revised manuscript.

  • Legend of figures is poor and barely understandable. Moreover, the figures and their relative legends should pair, whereas in the manuscript this is not the case.”
  • We corrected all figure legends, simplified and clarified them.

  • line numbering too often interferes with the Y axis of the figures thus making it difficult to read what is written on the axis and thus interpret and read plots.”
  • We corrected the readability of all figures.

  • it happens that in some figures (i.e. figure 2, figure 3) double-sided arrows appear but without any explanation in the legend. Quite odd? is it not?;.”
  • We added explanations for double sided arrows to the figure legends.

  • .5. “Extensive editing of English language and style required. Several oversights and typos are present throughout the main text. The language needs profound and extensive revisions because the manuscript is quite difficult to be read fluently.”.
  • We have revised the whole manuscript regarding English language and style as well as typos (see I).

  • Statistics require to be carefully revised because it seems that blunders are present, due to some kind of discrepancies between the plots and asterisks. Eventually, I warmly advise the authors to avoid using "tendentially significant", because in statistics either there is a significant difference or not. Trends are too weak to support data and thus enabling the authors to draw solid and robust conclusions.”
  • We revised the statistical explanations. Further, we deleted markings “tendentially significant”.

We hope, our manuscript is ready to be published in your journal now

Kind regards,

Helga Schmetzer

Round 2

Reviewer 2 Report

The authors have addressed scientific queries satisfactorily. However, stylistic errors still persist in the manuscript, especially the way the Introduction and Results sections are formatted. Entire Results is categorized under 3.1. which is incorrectly formatted. The authors must make changes to the formatting, in order to make the manuscript more easily readable.

Author Response

Dear reviewer 2,

First, we thank you for the (re)reviewing of our manuscript No. ijms-1825842.

I ‘General’. Please find attached all new revisions (highlighted in blue (in addition to preliminary revisions (labelled in yellow) including (further) English language/stylistic editing as well as shortening and reorganising/formatting of the manuscript.

With our statements below, we address all points of criticism:

  1. Response to Reviewer 2.

Please see also I, ‘GENERAL‘

-moderate English changes: we have revised

-some stylistic errors in the manuscript (esp. Introduction and Results): we have revised

-Reformatting and Reorganisation of Manuscript (esp. Results): We have reorganized the manuscript (especially Figure repositioning, Legend revision, the results‘ part).

We hope, our manuscript is ready to be published in your journal now

Kind regards,

Helga Schmetzer

Reviewer 3 Report

The quality of the manuscript has been improved.

Author Response

Dear reviewer 3,

First, we thank you for the (re)reviewing of our manuscript No. ijms-1825842.

I ‘General’. Please find attached all new revisions (highlighted in blue (in addition to preliminary revisions (labelled in yellow) including (further) English language/stylistic editing as well as shortening and reorganising/formatting of the manuscript.

With our statements below we address all points of criticism:

II. Response to Reviewer 3.

Please see also I, ‚GENERAL‘

-English language and style are fine/minor spell check required: we have revised

We hope, our manuscript is ready to be published in your journal now

Kind regards,

Helga Schmetzer

Reviewer 4 Report

I wish to thank the authors for the efforts made. Indeed, the revised version of the manuscript has improved when compared to the original. Nonetheless, the manuscript has not reached yet the standard required for a journal of IF above 6. The data are not fully convincing, and the weaknesses have only been partially fulfilled. One example among all: conclusions are not robustly supported by the data. In my experience, a correlation coefficient of 0.37 (fig 7 A) does not mean at all that the data are "(tendentially) correlated", thus difficult to claim that increased B7 integrin expression in CD3+ lymphocytes is associated with blast lysis. Minor concerns regard few typos, and why in the main text the authors start with table 2 instead that with table 1? It would be expected that when reading at first table 1 is encountered, and later table 2.

Author Response

Dear reviewer 4,

First, we thank you for the (re)reviewing of our manuscript No. ijms-1825842.

I ‘General’. Please find attached all new revisions (highlighted in blue (in addition to preliminary revisions (labelled in yellow) including (further) English language/stylistic editing as well as shortening and reorganising/formatting of the manuscript.

With our statements below we address all points of criticism:

II. Response to Reviewer 4.

Please see also I, ‚GENERAL‘

-English language and style are fine/minor spell check required: revisions done

-provision of more background information/references in the manuscript: we have included some more background information, as available: we have done several additional literature searches and did not find new literature in the field of our research

-improvement of research design and methods: we have clarified the design and description of some methods

-improvement of presentation of results/conclusions: revised : we have improved presentation of results , rewritten parts of the discussion and

 rewritten conclusions

-Position/citing of table 1 and 2 in the manuscript: table 1 and table 2 are revised and have a new position in the manuscript

Additionally, grammatical errors pointed out by the Academic Editor were fixed

We hope, our manuscript is ready to be published in your journal now

Kind regards,

Helga Schmetzer